# Comparison of Pro-Regenerative Effects of Carbohydrates and Protein Administrated by Shake and Non-Macro-Nutrient Matched Food Items on the Skeletal Muscle after Acute Endurance Exercise

**DOI:** 10.3390/nu11040744

**Published:** 2019-03-30

**Authors:** Eduard Isenmann, Franziska Blume, Daniel A. Bizjak, Vera Hundsdörfer, Sarah Pagano, Sebastian Schibrowski, Werner Simon, Lukas Schmandra, Patrick Diel

**Affiliations:** 1Institute for Cardiovascular Research and Sports Medicine, Department of Molecular and Cellular Sports Medicine, German Sports University, 50333 Cologne, Germany; e.isenmann@dshs-koeln.de (E.I.); franzi.blume@web.de (F.B.); D.Bizjak@dshs-koeln.de (D.A.B.); verahundsdoerfer@gmail.com (V.H.); sara.pagano@gmx.de (S.P.); lukas.schmandra@gmail.com (L.S.); 2Department of Fitness and Health, IST-University of Applied Sciences, 40233 Dusseldorf, Germany; 3Rheinische Fachhochschule Cologne, 50676 Cologne, Germany; sebastian.schibrowski@googlemail.com (S.S.); w.simon.314@web.de (W.S.)

**Keywords:** endurance exercise, skeletal muscle damage, inflammation, protein, carbohydrates, protein shake, food

## Abstract

Physical performance and regeneration after exercise is enhanced by the ingestion of proteins and carbohydrates. These nutrients are generally consumed by athletes via whey protein and glucose-based shakes. In this study, effects of protein and carbohydrate on skeletal muscle regeneration, given either by shake or by a meal, were compared. 35 subjects performed a 10 km run. After exercise, they ingested nothing (control), a protein/glucose shake (shake) or a combination of white bread and sour milk cheese (food) in a randomized cross over design. Serum glucose (*n* = 35), serum insulin (*n* = 35), serum creatine kinase (*n* = 15) and myoglobin (*n* = 15), hematologic parameters, cortisol (*n* = 35), inflammation markers (*n* = 27) and leg strength (*n* = 15) as a functional marker were measured. Insulin secretion was significantly stimulated by shake and food. In contrast, only shake resulted in an increase of blood glucose. Food resulted in a decrease of pro, and stimulation of anti-inflammatory serum markers. The exercise induced skeletal muscle damage, indicated by serum creatine kinase and myoglobin, and exercise induced loss of leg strength was decreased by shake and food. Our data indicate that uptake of protein and carbohydrate by shake or food reduces exercise induced skeletal muscle damage and has pro-regenerative effects.

## 1. Introduction

Consumption of proteins and carbohydrates after exercise via whey protein and glucose shakes is a common strategy for the enhancement of regeneration and physical performance after training [1]. There are reports that isolated amino acids, mainly leucine, can increase strength after exercise and result in a stimulation of the recovery of skeletal muscle after exercise [2]. Damage of muscle fibers by physical activity modulates protein synthesis, but simultaneously protein degradation [3]. Disturbance of the balance of muscle fiber degradation and synthesis leads to fiber degeneration and muscle atrophy, reduction of strength, and increase in muscle soreness [4,5].

Molecular mechanisms involved in muscle recovery include a modulation of protein synthesis and protein breakdown [6], an inhibition of the inflammatory response, and the activation of satellite cells [7]. Modulation of the balance of protein breakdown and protein synthesis is an important mechanism in skeletal muscle recovery and adaptation after exercise. It has been demonstrated that muscle protein breakdown rates are elevated in the days following resistance exercise [8]. In mouse knock out models, the inhibition of muscle protein breakdown impairs muscle recovery [9]. Some authors claim that modulation of protein breakdown is an important mechanism in muscle recovery in the period immediately after muscle damage [10]. In contrast, the stimulation of protein synthesis seems to be important for long term regeneration and hypertrophy. Therefore, all strategies affecting the balance between protein breakdown and synthesis in the period after damage will directly influence skeletal muscle recovery.

Protein metabolism in skeletal muscle has been shown to be stimulated by an ingestion of dietary proteins [11,12]. Whey protein supplementation results in a high availability of amino acids within the blood [13]. However, there are conflicting data about the effects of protein supplementation on physiological markers related to muscle recovery, like muscle strength, muscle soreness and serum creatine kinase (CK) concentrations after exercise [14,15]. These conflicting data are mainly caused by variables in the different studies. These variables include different amounts of nutritional feeding, timing, habitual food intake and type of exercise, volume of exercise, recovery measurements, and the timing of recovery measurement [16].

Apart from protein shakes, carbohydrate shakes are also frequently consumed to stimulate recovery after physical activity, containing protein/carbohydrates combinations. These drinks are recommended by the manufacturing companies to be ingested immediately after exercise, in order to have the most effective pro-regenerative effect. Research on this subject shows highly conflicting results. Some studies demonstrate that adding protein to carbohydrates improves the process of the recovery of endurance performance [17,18,19,20,21]. Others, focusing on the modulating of protein breakdown and synthesis, provide no evidence that carbohydrates increase exercise-induced protein secretion versus protein alone [22].

Mechanistically, it is believed that, beside the compensation of carbohydrate loss during exercise, carbohydrates also activate molecular mechanisms related to pro-regenerative effects. Here, insulin is claimed to be an important factor. It has been shown that a protein/carbohydrate combination leads to a higher increase in blood sugar and insulin concentration than just carbohydrate intake [23]. The molecular mechanism as to how proteins stimulate insulin secretion is unknown so far [24]. It also has been demonstrated that the amount of glycogen storage in skeletal muscle is higher with protein/carbohydrate combinations than with carbohydrates alone [23]. It is discussed that any uptake of carbohydrates immediately after exercise results in a strong increase of serum insulin, followed by a strong decrease of blood glucose. Binding of insulin to IGF-1 receptors should result in the activation of skeletal muscle specific signal transduction pathways, and via mTOR (Rapamycin), it should also result in a stimulation of protein synthesis in skeletal muscle [25,26]. Moreover, increasing insulin secretion also should stimulate amino acid uptake in skeletal muscle cells [26]. However, other investigations demonstrate that insulin does not stimulate muscle protein synthesis under physiological conditions in humans. It is very likely that the experimental design will have a huge impact on the results of the studies.

Beside ingestion of isolated amino acids, effects of protein/carbohydrate combinations on skeletal muscle could also be observed after the uptake of food containing protein and carbohydrates in a ratio of 70% to 30% [27]. In previous studies, we could observe pro-regenerative effects on the skeletal muscle after exercise, by eating combinations of dairy products and white bread [28]. Based on these observations, we concluded that eating food containing suitable concentrations of protein and carbohydrates may be an alternative strategy to promote skeletal muscle regeneration after exercise. Therefore, the aim of this follow up study was to compare the pro-regenerative effects of any protein/carbohydrate shake on the recovery of skeletal muscle after endurance exercise, with the ingestion of similar amounts of protein/carbohydrate combinations by eating a nearly iso-caloric meal.

Our hypothesis was that providing proteins/carbohydrate combinations will result in pro-regenerative effects, independent as to whether it was given by a shake or by eating a meal. Therefore, 35 nonspecifically trained subjects consumed either a whey protein/glucose shake, or a meal of white bread and sour milk cheese directly after endurance exercise. The following mechanistic read outs were determined: Serum glucose and serum insulin, serum creatine kinase (CK) and myoglobin (Myo) as muscle damage markers, hematologic parameters, cortisol, serum levels of the inflammation markers interleukin 6 (IL 6) and 10 (IL 10), as well as the macrophage migration inhibitory factor (MIF). In addition, leg strength as a functional marker for skeletal muscle regeneration was measured.

## 2. Methods

### 2.1. Participants

The study protocol has been approved by the local ethics committee (German Sports University, Cologne) and is in accord with the Declaration of Helsinki. It is registered in the German Clinical Trials Register under the title NUPROMU, and under the registration number DRKS-ID: DRKS00013359.

All participants provided written informed consent prior to their participation. The study excluded subjects who were currently taking any dietary supplements, sports drinks or functional food intended to enhance performance, or had taken any of these in the previous month. Moreover, subjects with known hypersensitivity to any of the constituents of the products under study (milk protein or lactose), were excluded. Throughout the study, subjects maintained their usual training routines and diets. Based on a statistical power analysis a total of 35 male participants (age: 23.2 ± 2.3 years; height: 181.5 ± 6 cm; weight: 77.9 ± 8.4 kg; mean ± standard deviation) were recruited for the study. Power was calculated with the program G *power (University of Düsseldorf, (Heinrich-Heine-Universität Düsseldorf (**HHU**)), Germany) based on effect sizes measured for the biological endpoints creatine kinase (CK) IL6 and IL10 in our pilot study [28]. Power analysis revealed that a number of 15 participants would be sufficient. Nevertheless, the group size was doubled, because additional parameters not measured in the pilot study were included. All participants were healthy and free of injury in the time period preceding the study. They were not specifically endurance trained, but they were physically active sports students. Anthropomorphic characteristics of the participants are indicated in Table 1.

### 2.2. Experimental Procedure

The aim of this study was to investigate the beneficial effects of a co-ingestion of protein and carbohydrate from a traditional food source in amateur sportsmen. For this purpose, 35 non-specific endurance trained male subjects performed a 10 km run with an intensity at 80% of their individual anaerobic threshold (IAT). Intensity was chosen based on previous investigations, demonstrating that 80% IAT and a distance of 10 km is a sufficient load for nonspecific endurance trained subjects to increase their serum CK [29]. The IAT was determined using an incremental field test with lactate determination according to a standard protocol [30]. Participants started with a speed of 2.5 m/s and distance of 800 m. After each successful termination of load level, the speed was increased by 0.5 m/s. The load duration was always between 5–6 min. Between the speed levels, 5 µL of capillary blood was taken from all participants. If a load level could not be successfully passed, the test was terminated.

The anaerobic threshold was used to calculate the meters per second and the resulting lap time in a track and field stadium (400 m lap length) for the 10 km run. Between the field test and the first investigation a wash out period of three weeks was used. Before running, the individuals were randomly divided into three intervention conditions—control, shake or food. Through a crossover design, each subject participated in each condition. However, the order was given after the first random assignment—control, shake, food.

Between the respective interventions there was a minimum wash out period of three weeks. The experimental design of the procedure at the intervention days is shown in Figure 1.

Blood samples were determined for different time points (Figure 1). In the morning, blood sample t0 was collected from the overnight fasted participants. Leg strength was tested by maximum three repetitions in back squat (3-RM BS) followed by s a defined small breakfast (Table 1).

30 min after breakfast, all participants started with the 10 km run with 80% of IAT. Immediately after exercise, subjects ingested either nothing (control), a combination of carbohydrates by eating white bread (35.3 g) and 100 g of a sour milk cheese high in protein (36.1 g) but very low in fat (3.5 g)(Loose GmbH, Leppersdorf, Germany), or by drinking a whey protein (52 g)/glucose (45 g) shake. The composition of the shake used was taken from a published study [21] where effects regarding skeletal muscle regeneration have already been demonstrated. It serves as a standard. The nutritional values of the foodstuffs used in each intervention are shown in Table 2. As a consequence of using whole foods, it is not possible to match the macronutrient content of the shake and foods exactly. Whole foods, like dairy products, usually contain fat. Nevertheless, we have opted for low-fat protein and carbohydrate sources as indicated in Table 2. In addition we have tried as accurately as possible to be isocaloric. For practical reasons our participants got the food as a sandwich composed of two slices of white bread, and 100g of sour milk cheese. So, we differed from the shake with regard to the calories around 17% (Table 2).

Immediately after ingestion, blood samples were determined at t1 (directly after exercise), followed by t.1.1 (+20 min after exercise), t1.2. (+40 min), t2 (+60 min), t3 (+120 h), t4 (+180 h) and t5 (+22 h). During the testing period, starting 12 h before exercise and 24 h after exercise, all participants were kept under a standardized nutrition to exclude additional effects by nutrition. All food for this 36 h period was provided to them.

12 h before exercise participants had a standardized dinner—spaghetti with tomato sauce (377 kcal, 15 g protein, 2.5 g fat, 72 g carbohydrates), and fasted until the standardized breakfast the next day. On the intervention day, participants were not allowed to eat until 3 h after exercise (t4; +180 min) except the provided nutritional compositions. After t4 they were allowed to eat the food provided in a period of 5 h after exercise until 8 pm. After 8 pm on the exercise day, participants fasted until the next day, when they were given the standardized breakfast. The standardized 24 h nutrition at the intervention day was calculated to fulfill all daily requirements regarding macro- and micronutrients, and provide sufficient calories.

Including the nutritive intervention, participants consumed a total of 3000 kcal a day. Daily macronutrient content was 146 g protein, 77 g fat and 411 g carbohydrates. Participants were allowed to drink water ad libitum.

## 3. Measurements

### 3.1. Determination of Serum Glucose and Serum Insulin

Samples were analyzed for glucose by oxidase method (COBAS Mira Plus; Roche Diagnostic Systems, Rotkreuz, Switzerland) and insulin by EIA (ALPCO diagnostics 1–2-3 Human Insulin EIA, Windham, NH, USA). Both serum concentrations were measured at all time points.

### 3.2. Determination of Serum Cortisol

Serum Cortisol concentrations were determined using the COBAS Mira Plus system (Roche Diagnostic Systems by ElectroChemiLumineszenz ImmunoAssay, ECLIA Rotkreuz, Switzerland). The serum cortisol concentration was measured at the time points t0, t1 (directly after exercise), t2 (+60 min), t3 (+120 min), t4 (+180 min) and t5 (+22 h).

### 3.3. Hematology

Blood samples were analyzed for routine CBC including total WBC count and differential for WBC sub fractions using a BC2300 hematology analyzer (Mindray Medical International Systems, Shenzhen, Peoples’ Republic of China). The Hematology parameters were determined at t0, t4 (+180 min) and t5 (+22 h).

### 3.4. Skeletal Muscle Creatine Kinase (CK mm) and Myoglobin (Myo)

Skeletal muscle specific creatine kinase activity (CKmm) and myoglobin (Myo) concentrations in the serum were determined using the COBAS h 232 Point-of-Care-System (Roche Diagnostic Systems, Rotkreuz, Switzerland) at t0 and t5 (+22 h).

### 3.5. Serum Cytokine Levels

IL-6 concentrations of serum samples were analyzed using the Human IL-6 ELISA Kit High. Sensitivity (Abcam, Cambridge, UK). IL10 serum concentrations were analyzed using a human IL-10. ELISA Kit (Abcam, Cambridge, UK). MIF serum concentrations were analyzed using a human MIF. ELISA Kit (Abcam, Cambridge, UK). Cytokine concentrations were determined at t0 and t5.

### 3.6. Leg Strength—3-RM Back Squat

The strength test was performed t0 and t5 (+22 h). The strength protocol is based on the guidelines of NSCA [31]. All subjects performed three warm-up sets and started at 50% of their planned maximum power with 10 repetitions. In the following warm-up sets, the weight was increased by about 10–20% and the repetition number reduced to five and three repetitions. This was followed by four 3-RM tests, starting with about 90% of the planned 3-RM. Between the sets a four-minute break was taken. The subsequent increases were made individually. If a load level was not successfully mastered twice in succession, the test was stopped.

### 3.7. Statistical Analyses

Quantitative variables were presented as mean values and standard deviations (SD). All measurement parameters were tested for the Normal Distribution. As a result, the Wilcoxon sign rank test (repeated measurement) and the Kruskal-Wallis test were used for Myo, CK, IL6, IL10, glucose and insulin. For MIF and 3-RM BS, a 2-way Analysis of Variance (ANOVA) with time and condition effects, and a dependent (Student’s) t-test, were used. The current version of SPSS (IBM SPSS Statistics 25.0, Ehningen, Germany) was used for statistical analysis, and *p* < 0.05 was taken as the level of statistical significance for all procedures. The images were created using GraphPad PRISM software (GraphPad Software, Inc. La Jolla, CA, USA).

## 4. Results

### 4.1. Effects of Exercise and Protein/Carbohydrate on Blood Glucose and Insulin Concentrations

It has been shown that the consumption of protein/carbohydrate combinations after exercise influences the blood sugar and insulin response, which is discussed to improve regeneration.

In Figure 2A, showing mean serum glucose concentrations of all participants (*n* = 35), a significant increase of serum glucose compared to t0 is detectable in the control and the shake condition at t1.1 (+20 min). A significant decrease compared to t0 is detectable in the shake condition and the food condition at t2 (+60 min).

In Figure 2B, mean serum insulin concentrations of all participants are shown. A significant increase of serum insulin compared to t0 is detectable in the food and the shake condition at t1.1 (+20 min), t1.2 (+40 min), t2 (+60 min) and t3 (+120 min).

### 4.2. Effects of Exercise and Protein/Carbohydrate on Hematopoietic Parameters

In agreement with published data [32] a significant decrease in the total leucocyte number in conditions could be observed 3 h (t4; +180 min) after exercise. This effect was not affected by any nutritive intervention (data not shown). After 24h (t5; +22 h), the total leucocyte number was again on the baseline in each condition. All other investigate hematopoietic parameters remained also unaffected by exercises and the nutritive intervention (data not shown).

### 4.3. Effects of Exercise and Protein/Carbohydrate on Blood Cortisol Levels

Serum Cortisol levels in athletes have been demonstrated to be influenced by physical activity and nutrition [33]. Figure 3 shows mean cortisol serum concentrations from all participants. The well described circadian rhythm of cortisol could be observed in all intervention conditions. Neither physical activity nor the nutritive interventions resulted in significant effects.

### 4.4. Effects of Exercise and Protein/Carbohydrate on Markers of Inflammation

Skeletal muscle damage results in an induction of inflammation. Interleukin 6 (IL 6), Interleukin 10 (IL 10), and Macrophage migration inhibitory factor (MIF) in the serum at the time point’s t0 and t4 (+180 min) were measured after exercise as markers for inflammation.

MIF serum levels (*n* = 18) were significantly increased in all conditions, however, in the food and shake conditions, the increase was significantly lower, compared to the control. (Figure 4A) IL 6 serum levels (*n* = 27) were significantly increased by exercise (Figure 4B). No significant differences could be observed between the shake and the food condition. Il10 serum levels (*n* = 28) were significantly increased in all conditions after exercise; however, the increase in the shake and food condition was significantly higher (Figure 4C).

### 4.5. Effects of Exercise and Protein/Carbohydrate on Serum Markers for Skeletal Muscle Damage

Skeletal muscle creatine kinase (CKmm) and myoglobin (Myo) are markers for skeletal muscle damage. Figure 5A shows the mean absolute Myo serum concentrations (*n* = 15) and Figure 5B shows the mean absolute of CKmm (*n* = 15).

Exercise in the control condition resulted in a significant increase of CKmm and Myo in the blood at t5 (+22 h). In the shake but also in the food condition exercise induced increase of skeletal muscle damage markers and was significantly lower compared to the control.

### 4.6. Effects of Exercise and Protein/Carbohydrate on Leg Strength as a Functional Marker for Skeletal Muscle Regeneration

We postulated that a better regeneration after endurance exercise will result in a lower loss of muscle strength (*n* = 15). As a functional approach for leg strength the 3-repetition maximum back squat was used. As seen in Figure 6, the uptake of protein/carbohydrate resulted in no significant loss of leg strength after endurance exercise. Only in the control condition did endurance exercise resulte in significantly lower leg strength at t5 (+22 h) after exercise.

## 5. Discussion

In the present study, we compared pro-regenerative effects of protein and carbohydrates administration on the skeletal muscle regeneration after exercise, given either by shake or by a standardized meal. Pro-regenerative effects of protein/carbohydrate combinations after endurance exercise have been described and mechanistically linked to a stimulation of insulin secretion [21]. Therefore, in this investigation blood glucose levels and insulin levels were measured at different time points after exercise and subsequent protein/carbohydrate ingestion.

It was clearly visible that exercise results in an increase of blood glucose concentrations 20 min after exercise termination. This is in agreement to our previous investigations [28] and confirms observations that moderate exercise increases blood glucose concentrations in healthy non-diabetic persons when a required exercise volume is achieved [34]. As expected, blood glucose concentration after administration of the shake extended a maximum 40 min after ingestion. In contrast, after food administration, no significant increase of blood glucose concentration could be observed, although a comparable amount of carbohydrates was ingested (Table 2). Both shake and also food ingestion resulted in a significant increase of insulin serum concentration, starting 20 min after exercise/ingestion, and reaching a maximum after 40 min. The increase of insulin in the food condition is lower, compared to a shake, which could be explained by the lower content of total carbohydrates (35 g compared to 45 g in the shake condition). Nevertheless, the kinetics of insulin in both conditions is absolutely comparable.

A possible explanation for the missing increase of blood sugar levels in the food condition may be the direct uptake of the resorbed sugars in the skeletal muscle and the liver. In the shake condition, same mechanisms may be relevant, but the faster and higher load of glucose could not be compensated completely. In general, our observation is supported by data demonstrating that, combining protein and carbohydrate increases insulin levels, but does not improve glucose response [35]. Our observation is also in line with published data, showing that the uptake of specific amounts of carbohydrates results in a much stronger stimulation of insulin secretion when they are combined with proteins [17,21].

Hematopoietic factors and cortisol serum concentrations were not affected in our study by any ingestion of carbohydrates and proteins, but the response of pro- and anti-inflammatory serum markers to exercise was strongly influenced. Here it has to be mentioned that the role of inflammation in the skeletal muscle’s adaptation to exercise is complex. Acute inflammatory response to exercise seems to promote skeletal muscle training adaptation and regeneration. In contrast, persistent, low-grade inflammation, as seen in a multitude of chronic diseases, is obviously detrimental [36]. Acute exercise has been described to induce an inflammatory response in the skeletal muscle. This can be detected by changes in the serum concentrations of inflammation-related cytokines [37,38]. Strenuous exercise results in changed serum concentrations of pro-inflammatory and anti-inflammatory cytokines like TNF alpha, IL-1, IL-6, IL-1 receptor antagonist, TNF receptors, IL-10, IL-8 and macrophage inflammatory protein-1 [38]. Increase of pro- and anti-inflammatory cytokines has been also described in animal models were skeletal muscle damage is induced by notoxin [39]. For these reasons the circulating levels of the pro-inflammatory cytokines IL6, and MIF and the anti-inflammatory cytokine IL10, were investigated 3 h after exercise in this study. We observed a significant increase of the serum concentrations of the pro-inflammatory cytokines IL6 and MIF after exercise in the control condition, which was significantly lower in both protein/carbohydrate conditions. In contrast, the serum concentrations of the anti-inflammatory cytokine IL10 were even more increased after exercise in both protein/carbohydrate conditions compared to the control condition. Remarkably, the effects after ingestion of the proteins by shake and by food are quantitatively and qualitatively comparable. These regulation patterns demonstrate that administration of protein/carbohydrate after exercise reduces the excretion of pro-inflammatory cytokines, but increases the excretion of anti-inflammatory proteins in the serum, which can be interpreted as an anti-inflammatory effect. Indeed, in an animal model where muscle damage was induced by administration of notoxin, accelerated skeletal muscle recovery correlated with increased IL10 expression and decreased TNF-alpha expression in the respective skeleton muscles [39]. As our study demonstrated, while a complete suppression of inflammation after exercise, e.g., after administration of glucocorticoids [40], promotes catabolic effects in the skeletal muscle, a modulation of the inflammatory response can be interpreted as an indication for pro-regenerative effects.

An important physiological endpoint investigated in this study was skeletal muscle damage and post-exercise recovery. Some studies show no effects of proteins/carbohydrates ingestion on skeletal muscle recovery [41] after exercise, while other investigations, however, have demonstrated a reduced post-exercise muscle soreness [42] and lower plasma concentrations of Myo [43] and CK [44] by administration of proteins/carbohydrates before, after, or during endurance exercise. Therefore, in our investigation CKmm and Myo concentrations were measured time-dependently after exercise, and a significant increase of the mean serum CK and Myo concentrations was detectable in the control condition 22h after training. The increase was not detectable in both protein/carbohydrate conditions. Our observation might be interpreted as an indication that skeletal muscle damage, induced by exercise, is lower after the uptake of protein/carbohydrate combinations. This is in line with the observation of Diel et al. [28] and previous described results [14]. Our data also indicate that administration of protein/carbohydrate combinations by food was as effective as administration by shake.

This hypothesis is further supported by our functional regeneration marker, the measurement of leg strength. The reduction of muscle strength especially after endurance training is well reported [45]. As molecular mechanisms for this reduction, reduced numbers of connective structures between actin and myosin filaments, and a reduced sensitivity for calcium, are discussed [46,47,48]. We postulate that a better regeneration after endurance exercise will result in a lower loss of muscle strength. As a functional approach, we have chosen the 3-repetition maximum back squat assay, which is believed to be one of the most effective training approaches based on the interaction of several muscle groups in the leg [49]. We found that the loss of leg strength after endurance is significantly lower in the nutritive intervention conditions, compared to the control condition. This is an obvious functional indication that uptake of protein/carbohydrate results in pro-regenerative effects. Again, food and shake are of comparable effectiveness.

Our study has several limitations. One limitation is that the food and the shake in our study do not match perfectly for macronutrient content and calories. However, for calories both groups differ only by 17%. With respect to macronutrient content, a perfect match is not possible because protein/carbohydrate shakes do not contain fat. Moreover, in our study, although there is no perfect match, food is lower in calories, protein and glucose compared to shake, but nevertheless we could observe comparable effects. Another limitation of our study is that important parameters related to skeletal muscle recovery, protein synthesis and protein breakdown, were not investigated. These are of course important endpoints which should be addressed in future studies and in a similar design.

Our study also has strength. As biological markers for skeletal muscle damage, we did not only focus on CK. CK is highly debated as a suitable parameter [50]. Therefore we have also analyzed Myo and used two independent markers for skeletal muscle damage in comparison. Looking for effects on the immune response, we have investigated a panel of different cytokines, as well pro- and anti-inflammatory ones. This also strengthens the interpretation of our results.

In summary, our results demonstrate that ingestion of protein and carbohydrate combinations by a shake, but also by a nearly isocaloric meal immediately after endurance exercise, affects a variety of physiological endpoints. In our study, uptake of protein and carbohydrate combinations by shake and meal resulted in a strong increase of insulin serum concentrations of the participants. Moreover, a modulation of any inflammatory response of the skeletal muscle towards exercise, indicated by reduced concentrations of pro-inflammatory markers, and an increase in anti-inflammatory markers, could be observed. The endurance training induced an increase of serum CKmm and Myo, markers of skeletal muscle damage, and as a functional marker for regeneration, the loss of leg strength was observed. All this could be antagonized by protein/carbohydrate combinations, given either by a shake or a meal, respectively. Mechanistically our data provide evidence that a combined uptake of protein and carbohydrates appears to reduce skeletal muscle damage after endurance exercise via a modulation of the immune response of the skeletal muscle. Also, insulin seems to be involved in initiation and mediation of the pro-regenerative effects. The most important part our observation is that all these beneficial effects can be achieved, either by the ingestion of food containing sufficient concentrations of carbohydrates and protein, or with the same efficiency as consuming a shake. This finding demonstrates again that it is possible to develop concepts to support training by a suitable diet without the need to consume nutrition supplements or isolated proteins.

## Figures and Tables

**Figure 1 nutrients-11-00744-f001:**
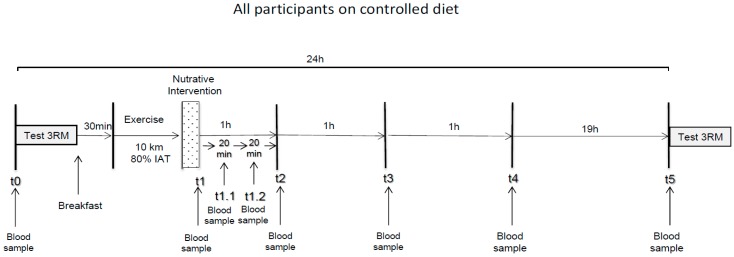
Experimental design of the study. IAT = individual anaerobic threshold. 3RM = leg strength.

**Figure 2 nutrients-11-00744-f002:**
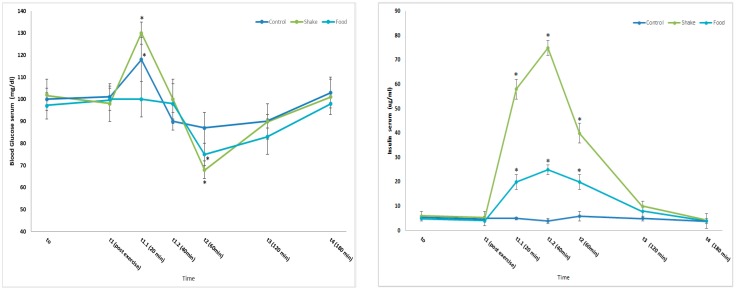
Effects of protein/carbohydrate uptake via food and shake, immediately after exercise, on serum glucose and serum insulin serum concentrations. Left: Mean serum blood glucose concentrations of all participants. All data are expressed as mean ± SD. * = *p* ≤ 0.05. Statistically significant differences between marked time points and respective t0 value are observed. Right: Serum insulin concentrations of all participants (Mean ± SD). * = *p* ≤ 0.01 show statistically significant differences between marked time point and respective t0 value in each condition.

**Figure 3 nutrients-11-00744-f003:**
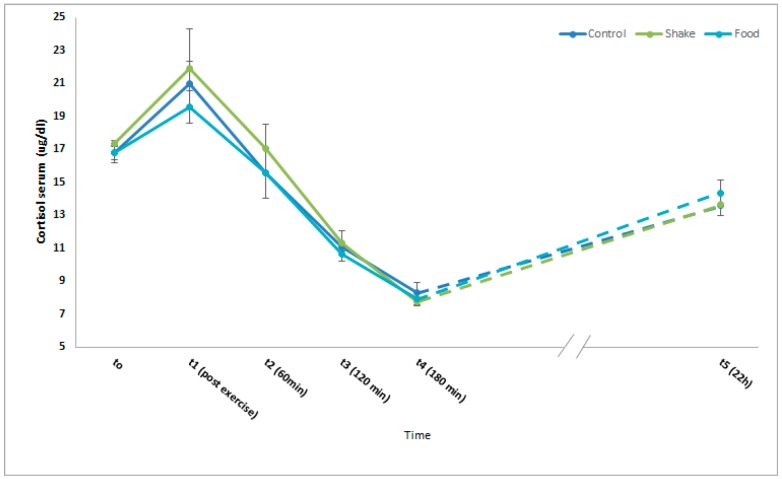
Effects of protein/carbohydrate uptake via food or shake, immediately after exercise, on serum cortisol concentrations. Mean serum cortisol concentrations of all participants. Mean ± SD. *p* ≤ 0.05 show statistically significant differences between marked group and respective t0 value.

**Figure 4 nutrients-11-00744-f004:**
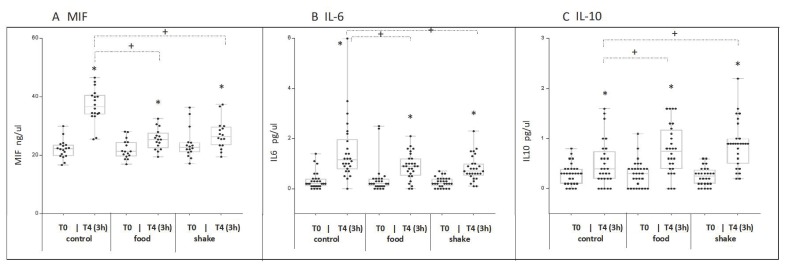
Effects of protein/carbohydrate uptake via food and shake, immediately after exercise, on serum levels of MIF (**A**) IL 6 (**B**) IL 10 (**C**) and 3h after exercise (t4; +180 min)). Shown are mean individual serum concentrations of the respective cytokines (A, B, C). Shown is mean ± SD. * = *p* ≤ 0.05, statistically significant differences between marked time point and respective t0 value. + = *p* ≤ 0.05 show statistically significant differences between marked groups.

**Figure 5 nutrients-11-00744-f005:**
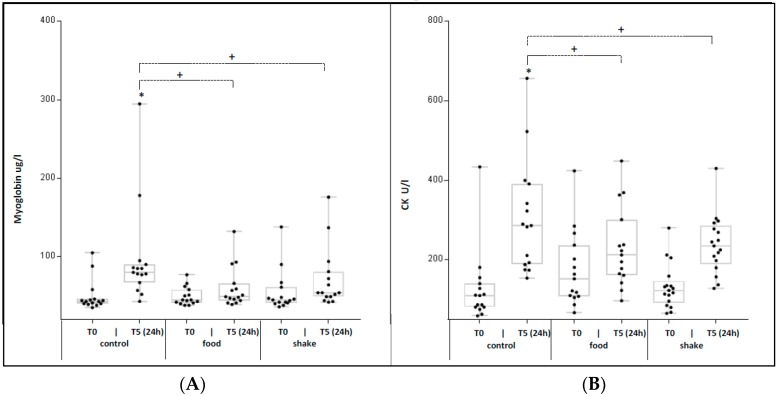
Effects of protein/carbohydrate uptake via food and shake on serum creatine kinase (CK) and myoglobin (Myo) concentrations, immediately after exercise, and 24 h after exercise. Shown are mean individual serum concentrations Myo (**A**) and CK (**B**). Mean of all individuals ± SD. * = *p* ≤ 0.05 shows statistically significant differences between marked time point and respective t0 value. + = *p* ≤ 0.05 show statistically significant differences between marked groups.

**Figure 6 nutrients-11-00744-f006:**
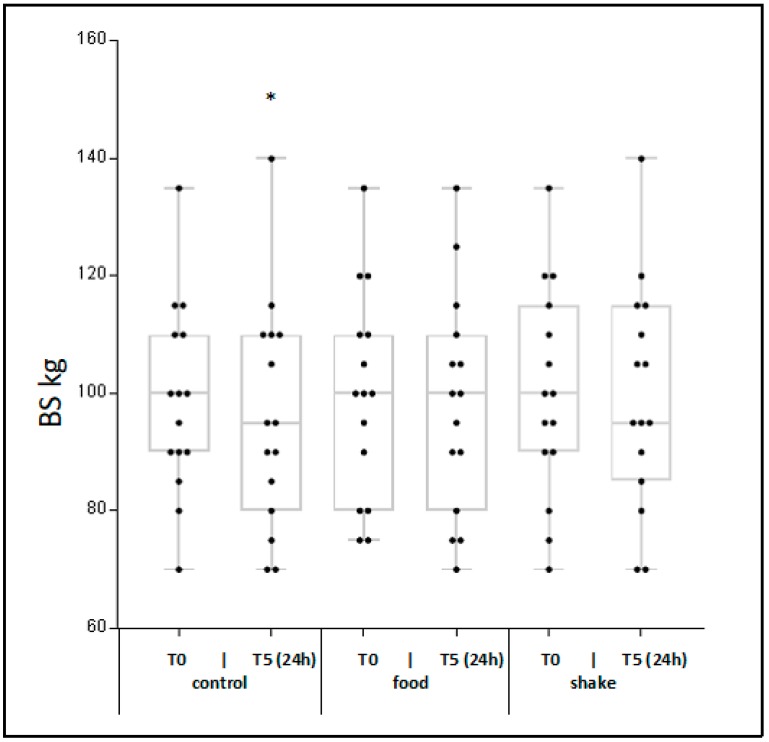
Effects of protein/carbohydrate uptake via food and shake on leg strength measured by the 3-repetition maximum back squat. In Figure 6 individual leg strength is shown Mean ± SD. * = *p* ≤ 0.01 show statistically significant differences between t5 and respective t0 value.

**Table 1 nutrients-11-00744-t001:** Compositions and calories of the standardized breakfast.

	Carbohydrates (g)	Protein (g)	Fat (g)	Calories (kcal)
30 g Cornflakes	25.2	2.1	0.3	111.9
250 mL milk (1.5% fat)	12.0	8.5	3.8	116.2
One banana (100 g)	27.4	1.3	0.4	118.4
Sum	64.6	11.9	4.5	346.5

**Table 2 nutrients-11-00744-t002:** Compositions and calories of protein/carbohydrate interventions.

		Carbohydrates (g)	Protein (g)	Fat (g)	Calories (kcal)
Food	76 g white bread and 100 g sour milk cheese	35.3	36.1	3.5	321
Shake	45 g Glucose 52 g Whey protein	45	52	0	386

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
