# Peer review of "Comparison of Pro-Regenerative Effects of Carbohydrates and Protein Administrated by Shake and Non-Macro-Nutrient Matched Food Items on the Skeletal Muscle after Acute Endurance Exercise"

_nutrients, 2019, doi:10.3390/nu11040744_

Reviewer 1 Report

Thank you for the kind invitation to review “Comparison of pro–regenerative effects of carbohydrates and protein administrated by shake or foodstuffs on the skeletal muscle after acute endurance exercise” by Isenmann et al. 

 Major points:

In the methods section the reviewers describe several analyses as standard laboratory tests, which appear to have been done by an external source/scientist. There’s simply not enough information supplied on the nature of these analyses, adequate enough for the experiments to be replicated.

 The standard of the graphs/figures is poor throughout, they are low resolution and poorly constructed, not appropriate for publication standard. Figure 2B and 2D are so small, and largely unintelligible.

 Figure 3B is not useful to the reader, it does not enhance the narrative in any way.

 Figure 4A-C are not needed, given the plots shown of that mean data in 4D-F – remove A-C. Shouldn’t the y-axis labels in D-F also be pg/ml? Since it’s still showing a measure of the levels of these cytokines.

 Again, Figure 5A-B are not needed, as per comments regarding Figure 4. Same for Figure 6.

 Line 420 – reference 33 is not an appropriate citation for the preceding statement.

Author Response

Answer to the referees.

We thank the referee for the helpful comments and tried our best to answer all questions and to revise the paper according to the comments of the referee.

 ·         In the methods section the reviewers describe several analyses as standard laboratory tests, which appear to have been done by an external source/scientist. There’s simply not enough information supplied on the nature of these analyses, adequate enough for the experiments to be replicated.

 We have modified the methods section and now provide detailed information regarding the used techniques and methods which will enable very interested reader of the paper to replicate the experiments.

·         The standard of the graphs/figures is poor throughout, they are low resolution and poorly constructed, not appropriate for publication standard. Figure 2B and 2D are so small, and largely unintelligible.

 ·         Figure 3B is not useful to the reader, it does not enhance the narrative in any way.

 ·         Figure 4A-C are not needed, given the plots shown of that mean data in 4D-F – remove A-C. Shouldn’t the y-axis labels in D-F also be pg/ml? Since it’s still showing a measure of the levels of these cytokines.

 ·         Again, Figure 5A-B are not needed, as per comments regarding Figure 4. Same for Figure 6.

 We have modified all graphs and enhanced resolution. We also tried to make the figures more accessible and took out all Figure parts mentioned by the referee.

·         Line 420 – reference 33 is not an appropriate citation for the preceding statement.

We have included a more relevant citation

Reviewer 2 Report

The authors have examined the impact of carbohydrates and protein provided in supplemental form or as whole-food items on markers of muscle recovery. Both the shake and food treatment improved markers of muscle recovery compared to the no feeding control treatment. Unfortunately, the interventions were not matched on macronutrient content: carbohydrate, protein, fat and total calories all differed between the food and shake treatment. This removes the main novelty of the study: a direct comparison of nutrients provided in supplemental or a whole-food matrix. Only indirect measurements of muscle recovery have been performed and only on a subset of all participants. The introduction and discussion are often superficial and include some incorrect statements. The writing of the manuscript needs to be improved, and the authors need to convince me that the current non-matched comparison is relevant, before I can recommend this manuscript for publication.

 Line 63-65. ‘’Molecular mechanism involved….a reduction in muscle damage’’

This sentence is not clear to me. Muscle protein breakdown rates are elevated in the days following resistance exercise [PMID: 9252485], which seems part of the adaptive response. Inhibition of protein breakdown in animal KO models impairs muscle recovery [PMID: 25380823]. While some inhibition of muscle protein breakdown may be beneficial, inhibition of muscle protein breakdown does not seem a key recovery mechanism [PMID: 30659499].

 Line 73-75. ‘’However, there are….concentrations after exercise’’

This sentence is not very informative. Please provide more information for the reader. What may be causing these conflicting results? Different amount of nutritional feeding, timing, habitual food intake, type of exercise, volume of exercise, recovery measurements, timing of recovery measurement? How does any of this relate to your research question?   

 Line 76-90: ‘These drinks…pro-regenerative effect’.

This statement is not referenced. In general, this section would benefit from more specific details. By what mechanism do carbohydrates improve recovery? Muscle glycogen repletion? The addition of carbohydrate to protein does not seem to impact muscle protein synthesis or muscle protein breakdown [PMID: 21131864]

 Line 86-92: ‘’In summary, these…..in skeletal muscle’’

This paragraph does not reflect the literature well. Insulin does not stimulate muscle protein synthesis rates under physiological conditions in humans [PMID:25646407].

 Line 123: ‘’Based on a power analysis a total of 35 male participants were recruited for the study’’.
Please provide full details of this power analyses. Including: what was the primary outcome of this study for which the power analyses was calculated? On which data was the power calculation based?

 Table 3. Composition of…interventions
Why were the treatments not macro-nutrient matched to allow to compare the impact of food matrix on recovery (ostensibly the main novelty of this study)?

 Line 127: ‘’in table 1’’

The manuscript that I have received does not seem to have a Table 1 included?

 Line 170: ‘’t1”

Please write these time points in relative time point of your intervention throughout the manuscript. I.e. t1 would be (t= 20 min).

 Line 387: ‘A possible explanation…and the liver’.

Can the authors support this argument with literature? Has a similar concept seen before where carbohydrate feeding results in lower postprandial glucose levels compared to not feeding?

 Figure 2 (and others)

Please adjust the axis to be proportional to the difference in time between the time point. That is, the distance between t1 and t2 should be the same as the distance between t2 and t3. Please remove the horizontal lines (other than the X-axis). Fig 2A: Please better indicate that the origin does not start at zero.

 Fig 4 (and others)

Why were not all 35 subjects analyzed?

 Line 396 ‘The response of…was strongly influenced’.

Is needs to be better discussed if this a positive response? [PMID: 25826388]

 Discussion
There current study has several limitations. It would be useful to discuss these in a paragraph. In addition to my previous comments, is creatine kinase a reliable indication of muscle damage [PMID: 22288008]?

 Language
The manuscript needs to be carefully proofread for the resubmission.

Author Response

Answer to the referees.

We thank the referee for the helpful comments and tried our best to answer all questions and to revise the paper according to the comments of the referee.

1.     The authors have examined the impact of carbohydrates and protein provided in supplemental form or as whole-food items on markers of muscle recovery. Both the shake and food treatment improved markers of muscle recovery compared to the no feeding control treatment. Unfortunately, the interventions were not matched on macronutrient content: carbohydrate, protein, fat and total calories all differed between the food and shake treatment. This removes the main novelty of the study: a direct comparison of nutrients provided in supplemental or a whole-food matrix.

We thank the referee for this comment.  The main focus of this study to compare whether carbohydrates and protein provided in supplemental form or as whole-food differ in their effects regarding skeletal muscle regeneration after exercise. This is also the reason why carbohydrate, protein, fat and total calories all differ between the food and shake treatment. The composition of the  shake was taken from a published study where effects regarding skeletal muscle regeneration have been already demonstrated. It serves as standard. Using whole food means that it is not possible to match exactly regarding the macronutrient content of the shake at all, because in contrast to the shake whole food always contains fat. We have chosen low fat protein and carbohydrates sources, nevertheless like indicated in Tab. 2 they contain fat. Therefore, we have tried to be as close as possible isocaloric.  For practical reasons our volunteers get the food in a form of a sandwich composed of two slices of white bread and 100 g of sour milk cheese. So we differ with regard to the calories around 17% from shake (lower): We feel this is acceptable to claim it nearly isocaloric. With respect to the content of protein and carbohydrates we are lower in food than in shake. However getting to the literature it turns out that the amount of protein and carbohydrates differ from study to study. Important in our opinion is that ever we are lower compared to shake our effects are nearly comparable for the measured end points.  We have tried to modify the introduction, the methods section and the discussion to include these issues into the paper.

2.     Only indirect measurements of muscle recovery have been performed and only on a subset of all participants.

Here we think it is important to discuss what are suitable direct parameters operationalizes to skeletal muscle regeneration. We thank the referee for the links to relevant and important papers. It seems that the referee mainly operationalize stimulation of protein synthesis to skeletal muscle recovery. We agree that this is an important biological parameter which we unfortunately don’t measure. However from a functional point of view  the most relevant parameter for skeletal muscle recovery it is its ability to produce power. Therefore that’s our main end point for functional skeletal muscle regeneration we measured. We did it only in a smaller subset of the participants because this functional test takes a lot of time and it was not possible to include it in the experimental design for all participants for capacity reasons. However here it is important to point out that the number of participants was chosen based on a power analysis. This power analysis based on the effect size observed in the mentioned pilot study for the biomarkers CK and inflammation. In the power analysis we end up with a number of 15 participants’. Nevertheless we decide to include 35 for the analysis of immune response and markers for skeletal muscle damage.

 3.     Line 63-65. ‘’Molecular mechanism involved….a reduction in muscle damage’’

This sentence is not clear to me. Muscle protein breakdown rates are elevated in the days following resistance exercise [PMID: 9252485], which seems part of the adaptive response. Inhibition of protein breakdown in animal KO models impairs muscle recovery [PMID: 25380823]. While some inhibition of muscle protein breakdown may be beneficial, inhibition of muscle protein breakdown does not seem a key recovery mechanism [PMID: 30659499].

We thank the referee for this helpful comment and agree that the contend of this text passage needs to be discussed more differentiated. So we have modified it.

 4.     Line 73-75. ‘’However, there are….concentrations after exercise’’

This sentence is not very informative. Please provide more information for the reader. What may be causing these conflicting results? Different amount of nutritional feeding, timing, habitual food intake, type of exercise, volume of exercise, recovery measurements, timing of recovery measurement? How does any of this relate to your research question?

  We modified this passage and also come back to this issue in the discussion section of our paper

 5.     Line 76-90: ‘These drinks…pro-regenerative effect’.

This statement is not referenced. In general, this section would benefit from more specific details. By what mechanism do carbohydrates improve recovery? Muscle glycogen repletion? The addition of carbohydrate to protein does not seem to impact muscle protein synthesis or muscle protein breakdown [PMID: 21131864]

We have modified this passage and argue now more differentiated. We also include the reference provided by the referee

 6.     Line 86-92: ‘’In summary, these…..in skeletal muscle’’

This paragraph does not reflect the literature well. Insulin does not stimulate muscle protein synthesis rates under physiological conditions in humans [PMID:25646407].

We have modified this passage of the introduction and include the reference provided by the referee, again to demonstrate that there are confliction results.

 7.     Line 123: ‘’Based on a power analysis a total of 35 male participants were recruited for the study’’.
Please provide full details of this power analyses. Including: what was the primary outcome of this study for which the power analyses was calculated? On which data was the power calculation based?

 We have added a paragraph regarding the power analysis in the method section including of a citation refereeing to the pilot study on which the power analysis is based.

8.     Table 3. Composition of…interventions
Why were the treatments not macro-nutrient matched to allow to compare the impact of food matrix on recovery (ostensibly the main novelty of this study)?

 This point we have already addressed in the introduction of our answers to the referee. The composition of the used shake was taken from a published study (liT) where effects regarding skeletal muscle regeneration have been already demonstrated. It serves as standard. Using whole food means that it is not possible to match exactly regarding the macronutrient content of the shake at all, because in contrast to the shake whole food always contains fat. We have chosen low fat protein and carbohydrates sources, nevertheless like indicated in Tab. 2 they contain fat. Therefore we have tried to be as close as possible is caloric. For practical reasons our volunteers get the food in a form of a sandwich composed of two slices of white bread and 100 g of sour milk cheese. So we differ with regard to the calories around 17% from shake (lower): We feel this is acceptable to claim it nearly isocaloric. With respect to the content of protein and carbohydrates we are lower in food than in shake. However getting to the literature it turns out that the amount of protein and carbohydrates differ from study to study. Important in our opinion is that ever we are lower compared to shake our effects are nearly comparable for the measured end points. We have tried to modify the introduction the methods section and the discussion to include these issues into the paper.

 9.     Line 127: ‘’in table 1’’

The manuscript that I have received does not seem to have a Table 1 included?

We have corrected

 10.  Line 170: ‘’t1”

Please write these time points in relative time point of your intervention throughout the manuscript. I.e. t1 would be (t= 20 min).

We modified this in the paper

 11.  Line 387: ‘A possible explanation…and the liver’.

Can the authors support this argument with literature? Has a similar concept seen before where carbohydrate feeding results in lower postprandial glucose levels compared to not feeding?

We have checked, but we have not found published studies in a similar design, which have paid attention to this aspect. In our pilot study we have also not seen this effect because we did not check after 20 min and 40 min. If you only check after 1 h you will miss it.

  12.  Figure 2 (and others)

Please adjust the axis to be proportional to the difference in time between the time point. That is, the distance between t1 and t2 should be the same as the distance between t2 and t3. Please remove the horizontal lines (other than the X-axis). Fig 2A: Please better indicate that the origin does not start at zero.

We have modified Fig.2 and 3 . According to the suggestion of the referee we take out the horizontal lines and better indicate that the origin does not start at zero by taking out the lines on the X-axis). We did not adjust the axis to be proportional to the difference in time between the time point. If we do this, especially for the glucose and Insulin data the figures shrink to much in the first hour and we lose resolution.

 Fig 4 (and others)

13.  Why were not all 35 subjects analyzed?

For blood glucose, insulin and cortisol (Fig. 2 and 3) we did analyses nearly a time points of all volunteers. Based on the comment of the other referees her we took out the figures showing individual data. The parameter leg strength (Fig. 6) indeed was only analyzed for a subset of individuals. Here we did it only in a smaller subset of the participants because this functional test takes a lot of time and it was not possible to include it in the experimental design for all participants for capacity reasons. For the cytokines, Myo and CK only data set were shown and considered for statistical analysis which were complete. For technical reasons not in all 35 individuals all parameters could be successfully detected for all time points. Sometimes Elisa does not work or was under detection limit sometimes blood samples were not properly processed and get lost. So for these parameters only data from those individuals were analyzed which were available for all time points in all interventions.

 14.  Line 396 ‘The response of…was strongly influenced’.

Is needs to be better discussed if this a positive response? [PMID: 25826388]

We referred to the mentioned paper and improved the discussion of this paragraph according the suggestions of the referee (reference to effects of Glucocorticoids…)

 Discussion
There current study has several limitations. It would be useful to discuss these in a paragraph. In addition to my previous comments, is creatine kinase a reliable indication of muscle damage [PMID: 22288008]?

We added a paragraph in the discussion regarding the limitations

 Language
The manuscript needs to be carefully proofread for the resubmission.

We tried our best to do this

Round  2

Reviewer 1 Report

I have no further comments.

Author Response

We thank the reviewer

Reviewer 2 Report

 The authors have done well to improve the manuscript. However, there are still a few points that require attention.
Title: The authors have provided rationale for why the shake and food items are not macro-nutrient matched. The current title suggests that the exact same nutritional intervention is given, just as a shake or food items. If the title is rephrased, that would away my concern.
The power calculation is problematic. A power calculation should be made for the main outcome, which was skeletal muscle recovery. A power calculation should not be based on different endpoints and/or multiple endpoints (i.e CK, IL6, IL10). A study should also not include more subjects than for the primary outcome. I strongly recommend the authors to pay better attention to their power calculation in future work, as the different number of subjects for each analyses makes the manuscript seem chaotic. The abstract should indicate that not all outcomes were measured in all 35 subjects: e.g. put n=x after each outcome.
  Figure 2 (and others). Please adjust the axis to be proportional to the difference in time between the time point. That is, the distance between t1 and t2 should be the same as the distance between t2 and t3. Please remove the horizontal lines (other than the X-axis). Fig 2A: Please better indicate that the origin does not start at zero.

We have modified Fig.2 and 3 . According to the suggestion of the referee we take out the horizontal lines and better indicate that the origin does not start at zero by taking out the lines on the X-axis). We did not adjust the axis to be proportional to the difference in time between the time point. If we do this, especially for the glucose and Insulin data the figures shrink to much in the first hour and we lose resolution.

Fig 2 and 3 have been improved. However, the axis should really be made proportional to the time. This can easily be done by improving the size of the whole figures and placing the figures below each other, instead of next to each other.  

Author Response

Answers to the referee

We want to thank the author for his further comments and tried our best to modify the paper  based on the sugestions.

1.     Title: The authors have provided rationale for why the shake and food items are not macro-nutrient matched. The current title suggests that the exact same nutritional intervention is given, just as a shake or food items. If the title is rephrased, that would away my concern. 

We have changed the title of the paper according the suggestions of the referee

2.     The power calculation is problematic. A power calculation should be made for the main outcome, which was skeletal muscle recovery. A power calculation should not be based on different endpoints and/or multiple endpoints (i.e CK, IL6, IL10). A study should also not include more subjects than for the primary outcome. I strongly recommend the authors to pay better attention to their power calculation in future work, as the different number of subjects for each analysis makes the manuscript seem chaotic. The abstract should indicate that not all outcomes were measured in all 35 subjects: e.g. put n=x after each outcome. 

We thank there referee for this explanation. Indeed power calculation was based on the endpoint CK. The more relevant parameter leg strength was not tested in the pilot study. Therefore we had no idea regarding the effect size of this parameter. We have indicated in the abstract  like suggested by the referee that not all outcomes were measured for all 35 subjects,

3.     Figure 2 (and others). Please adjust the axis to be proportional to the difference in time between the time point. That is, the distance between t1 and t2 should be the same as the distance between t2 and t3. Please remove the horizontal lines (other than the X-axis). Fig 2A: Please better indicate that the origin does not start at zero.

We have modified Fig.2 and 3 . According to the suggestion of the referee we take out the horizontal lines and better indicate that the origin does not start at zero by taking out the lines on the X-axis). We did not adjust the axis to be proportional to the difference in time between the time point. If we do this, especially for the glucose and Insulin data the figures shrink to much in the first hour and we lose resolution.

Fig 2 and 3 have been improved. However, the axis should really be made proportional to the time. This can easily be done by improving the size of the whole figures and placing the figures below each other, instead of next to each other. 

 We have modified Fig.2 and 3 now according to the suggestion of the referee.